# Evaluation of Maximum Entropy (Maxent) Machine Learning Model to Assess Relationships between Climate and Corn Suitability

Abigail Fitzgibbon [1,*], Dan Pisut [1]ⓘ and David Fleisher [2]ⓘ

1   Esri, Redlands, CA 92373, USA
2   Adaptive Cropping Systems Laboratory, United States Department of Agriculture, Agricultural Research Service, Beltsville, MD 20705, USA
*   Correspondence: afitzgibbon@esri.com

**Abstract:** Given the impact that climate change is projected to have on agriculture, it is essential to understand the mechanisms and conditions that drive agricultural land suitability. However, existing literature does not provide sufficient guidance on the best modeling methodology to study crop suitability, and there is even less research on how to evaluate the accuracy of such models. Further, studies have yet to demonstrate the use of the Maximum Entropy (Maxent) model in predicting presence and yield of large-scale field crops in the United States. In this study, we investigate the application of the Maxent model to predict crop suitability and present novel methods of evaluating its predictive ability. Maxent is a correlative machine learning model often used to predict cropland suitability. In this study, we used Maxent to model land suitability for corn production in the contiguous United States under current bioclimatic conditions. We developed methods for evaluating Maxent's predictive ability through three comparisons: (i) classification of suitable land units and comparison of results with another similar species distribution model (Random Forest Classification), (ii) comparison of output response curves with existing literature on corn suitability thresholds, and (iii) with correlation of predicted suitability with observed extent and yield. We determined that Maxent was superior to Random Forest, especially in its modeling of areas in which land was likely suitable for corn but was not currently associated with observed corn presence. We also determined that Maxent's predictions correlated strongly with observed yield statistics and were consistent with existing literature regarding the range of bioclimatic variable values associated with suitable production conditions for corn. We concluded that Maxent was an effective method for modeling current cropland suitability and could be applied to broader issues of agriculture–climate relationships.

**Keywords:** Maxent; machine learning; crops; climate; corn; suitability; species distribution; agriculture

## 1. Introduction

Global climate change has impacted the contiguous United States (CONUS), affecting agricultural indicators such as average growing season temperature [1]. Since 1991, hotter summers, changes in precipitation, and increasingly frequent extreme weather events have accounted for approximately $27 billion in insured agricultural losses [2]. Moreover, with the climate projected to warm over the coming decades, risks to agriculture will likely increase [3,4]. These risks create an urgency to model how agriculture will fare under future climate conditions. Doing so, however, is a difficult task that depends on a variety of atmospheric, agronomic, and economic factors [5]. The first step in such an analysis is an evidence-based method for understanding current cropland suitability. With a solid grasp of the environmental mechanisms driving current suitability, future suitability can be anticipated. To do so, solid methodology must be developed and carefully evaluated to

ensure that it can accurately and efficiently predict species presence and abundance (yield) with the ability for future predictions. In this paper, we devise a geographic approach to understanding crop suitability using machine learning to advance the capability of evaluating current and future agriculture–climate relationships.

Land suitability analysis (LSA) is a quantitative approach to evaluate the ability of a predetermined area to perform an ecological task (e.g., growing crops) given multiple environmental criteria [6,7]. LSA has been greatly facilitated by the emergence of increasingly powerful geographic information systems (GIS), expediting analysis and visualization of multiple environmental factors [8]. Using GIS, one of the most common approaches to studying land suitability is the analytical hierarchy process (AHP) [9–11]. AHP, or multiple-criteria decision analysis works by scoring or weighing criteria and adding them together to create a composite score. These types of multiple criteria analysis LSAs have been used to make predictions about crop yield [12,13]. However, unlike more statistically rigorous processes, this method relies on expert opinion to dictate how scores are assigned and criteria weighed [14], which makes APH implicitly subjective and somewhat imprecise [15]. Moreover, user-defined methods such as AHP are meant to study current suitability under current environmental conditions. However, studying the impact of climate change involves predicting using non-analogous climates, making it an inherently extrapolative process that an AHP is not meant to handle [16–18]. There are very few analytical hierarchy or fuzzy logic LSA studies that apply future climate conditions [19]. Therefore, AHP is an insufficient method for predicting to future climates, and a more rigorous and thorough method is needed to confront the challenge of extrapolative predictions to current and future conditions.

We employ a more complex type of model classified as a type of species distribution model (SDMs), to assess cropland suitability. SDMs apply machine learning to make predictions, making them more capable of extrapolation. Extrapolation—either to broader study areas or non-analogous climates—has become the primary application for many SDMs [20]. SDMs use numerical methods to relate an observed presence or abundance to environmental conditions [21]. SDMs are mostly known for their applications in conserving endangered or otherwise unique flora and fauna [22] and fall into two general categories: correlative and mechanistic. The two differ in that mechanistic models are programed with explicit species-environment relationships and predict based on those given conditions. In contrast, correlative models identify those relationships by analyzing patterns between presence and environmental conditions [22]. Mechanistic models (also known as process-based models) generally produce accurate results in modeling species distribution [23,24]. Mechanistic models are also well suited to extrapolating to future climates since they are given explicit environmental relationships [25]. Results derived from mechanistic models have been shown to demonstrate strong correlation between simulated and observed corn yields at a regional level under current conditions [26]. However, it is unclear whether mechanistic models truly outperform correlative in prediction to future climates [21]. In fact, there are several distinct disadvantages of mechanistic SDMs. Mechanistic models need extensive amounts of data and intensive time and processing requirements, which makes them ill-suited for large-scale predictions to vast geographic areas [23,27]. The data needed for mechanistic models generally require costly, long-term observational or experimental studies that are not available for all species [24]. For this reason, correlative models may be preferable when the relationship between species presence and environmental conditions is unknown, highly complex, or over a large geographic scale. Further, a correlative SDMs that is able to accurately predict crop extent and yield—without the data, time, and processing cost involved in mechanistic models—would demonstrate the utility and relative ease of such an approach and have broad applications to agricultural modeling generally.

Without explicit mechanistic relationships, correlative SDMs rely on a set of environmental characteristics and species distribution data to identify relationships between the environment and species presence. For most crop suitability analyses, explanatory variables consist of climatic, soil, or topographic variables. Together, these three types of data

provide a well-rounded picture of environmental conditions and constitute the essential datasets for suitability assessments [9–12,28–32]. Bioclimatic data such as temperature and precipitation can significantly affect photosynthetic efficiency and growing season duration [28,33]. Topographic variables include slope and elevation and are also generally essential to suitability [30]. Soil drainage, root zone depth, and available water storage are also critical limiting factors in crop suitability [30,34,35]. Correlative models take these site conditions and identify ranges and thresholds where a given species is likely to be found.

The Maximum Entropy model (Maxent) is a correlative machine learning model with broad applications in ecology and suitability studies [22,36,37]. Maxent generates a correlation between values of explanatory variables and locations of known species presence [38]. Maxent is known for its predictive skill and its ability to function even with low sample sizes and lack of absence points [36]. Though most SDMs are applied to conservation biology, Maxent is uniquely suited to study climate-sensitive crops [39]. It also handles extrapolation well, which helps predict crop suitability as predictions will inevitably need to be made outside of areas of presence [16–18]. Maxent is increasingly used in cropland suitability assessments, particularly with respect to climate [30]. Other correlative methods applied to predicting crop suitability include Random Forest, Boosted Regression Tree, Generalized Linear Model, and others [40]. There are a few established methods of evaluating such models. The most common is the use of scalar classification metrics and confusion matrices, particularly including metrics such as accuracy, sensitivity, and recall [23,27]. The latter two metrics may be evaluated within a receiver operator curve (ROC) [40]. Another less common method of evaluation is correlation with observed crop extent and yield statistics [23,26]. Few studies, if any, combine both of these evaluation methods.

Corn (*Zea mays* L.) is a major commodity crop and critical component of global food security. While cultivated widely, its main limiting factors to production are defined by bioclimatic conditions [41]. By area, corn is the most cultivated crop in the CONUS [42]. In this paper, we analyzed the ability of Maxent to accurately predict and provide meaningful insight into corn suitability in the CONUS. An accurate Maxent model will be able to identify the relationship between these conditions and corn presence and generate a map of suitable areas. Maxent was evaluated via comparison against another frequently used machine learning-based SDM (Random Forest), comparison with independent yield and harvest area data, and in the context of existing literature on bioclimatic suitability conditions. We evaluated Maxent's predictive ability using multiple metric methods to develop a thorough framework for model evaluation. We hypothesized that Maxent would outperform a Random Forest (RF) approach and generate accurate predictions when compared to independent production statistics and existing literature.

The purpose of this manuscript is thus to (i) demonstrate novel methods in how Maxent can be implemented to study U.S. corn suitability and (ii) introduce new methods to evaluate Maxent's predictive ability. This manuscript is organized as follows: We begin with showing the materials and methodology followed to produce corn suitability results using Maxent. Next, we introduce a new set of methods for evaluating Maxent, including comparison with Random Forest using scalar classification metrics, investigation of response curves, and correlation of predicted and observed extent and yield. Upon determination of Maxent's utility and accuracy in predicting corn suitability, we then use Maxent in practice to derive further insights into suitable conditions and risks. Maxent's success in this study would demonstrate that this model is particularly appropriate for cropland suitability assessments. Furthermore, solid evidence of Maxent's application to cropland suitability will lend credibility to future studies looking to apply machine learning to future climate scenarios and evaluate the risk climate change poses to agriculture. In developing a methodology not only for running but also evaluating Maxent, we add to the body of literature on correlative machine learning and provide novel insights into the field of agricultural modeling.

## 2. Materials and Methods

### 2.1. Data

2.1.1. Cropland Presence Data and Study Area

The United States Department of Agriculture (USDA) Cropland Data Layer was used to train the model for crop presence. It includes detailed records of 133 crop types at 30-m resolution with full CONUS coverage since 2008. The georeferenced raster is derived from satellite imagery with extensive ground-truthing, making it a highly accurate assessment of current crop presence and absence [42]. The predominant crop type for each pixel was determined for the five years from 2016 to 2020. The resulting raster was classified into corn and non-corn cropland pixels.

We excluded, both in training and prediction, all non-cropland, including developed areas, forest, open water, shrub, pasture, and all other lands not predominantly engaged in crop cultivation from 2016 to 2020. Non-cropland was excluded because it is inherently unsuitable for large-scale agriculture since it is used for other purposes (urban areas, pasture) or incapable of supporting crops (open water, ice). Given established trends of decreasing US farmland, it is very unlikely that non-cropland will be converted to cropland [43]. Exclusion of non-cropland likely improves the model in two ways: time and accuracy: (1) the model runs much more efficiently when it does not have to predict to areas that would be unsuitable for large-scale agriculture anyway, and (2) differences in environmental conditions within cropland suitable for different crops (growing corn vs. growing soy) are more subtle than those between cropland and non-cropland (growing corn vs. perennial snow/ice). Therefore, the model must perform more sophisticated, nuanced analysis to parse difference and identify suitability within cropland-only areas.

2.1.2. Factors Driving Crop Suitability

A set of 19 bioclimatic variables downloaded from WorldClim [3] provide average climate conditions for 1970–2000 at 30 arc-s resolution (Table 1). WorldClim data is routinely used for cropland suitability studies because they provide a comprehensive picture of monthly, quarterly, and annual bioclimatic conditions [35,44,45]. The 1-km rasters were resampled to a 180-m resolution.

In addition to the 19 bioclimatic variables, we selected a limited number of fundamental soil and topographic variables, including slope, elevation, root zone depth, and available water storage based on the aforementioned literature (Table 1). We also selected soil taxonomic order to account for other soil variables concurrently. The USDA National Cooperative Soil Survey (NCSS) classifies 12 orders of soils based on characteristics including depth, moisture, temperature, texture, structure, cation exchange capacity, base saturation, clay mineralogy, organic matter content, and salt content [46]. As a categorical variable encompassing multiple conditions, taxonomic order does the work of several individual variables. Soil variables were sourced from the USDA NCSS, which offers 10-m resolution soil data. To ensure that no area is left with NULL data values (indicating a lack of soil data), the soil rasters were resampled and smoothed to 180-m resolution using focal statistics.

**Table 1.** Twenty-four variables incorporated in the Maxent and Random Forest models. Nineteen bioclimatic variables were derived from WorldClim and included historic and future predictions based on representative greenhouse concentration pathway scenario (RCP) 4.5 [3]. The U.S. Geological Survey (USGS) provides interpretation and calculations for each bioclimatic variable [47]. The five soil and topographic variables are derived from the National Soil Cooperative Survey [46].

| Variable | Definition |
| --- | --- |
| Bio 01: Mean annual temperature | Annual Mean temperature |
| Bio 02: Mean diurnal range | Average difference between high and low daily temperature |

**Table 1.** *Cont.*

| Variable | Definition |
| --- | --- |
| Bio 03: Isothermality | Ratio of mean diurnal temperature range relative to seasonal range |
| Bio 04: Temperature seasonality | Temperature variation over a year by monthly average temperature |
| Bio 05: Max temp. of warmest month | Monthly mean of daily high temperatures for hottest month |
| Bio 06: Min temp. of coldest month | Monthly mean of daily low temperatures for coldest month |
| Bio 07: Temperature annual range | Bio 07 = Bio 05–Bio 06 |
| Bio 08: Mean temp. of wettest quarter | Average temperature for three months with most precipitation |
| Bio 09: Mean temp. of driest quarter | Average temperature for three months with least precipitation |
| Bio 10: Mean temp. of warmest quarter | Average temperature for three hottest months |
| Bio 11: Mean temp. of coldest quarter | Average temperature for three coldest months |
| Bio 12: Annual precipitation | Total annual precipitation |
| Bio 13: Precipitation of wettest month | Total precipitation for month with most precipitation |
| Bio 14: Precipitation of driest month | Total precipitation for month with least precipitation |
| Bio 15: Precipitation seasonality | Precipitation variation over a year by monthly total precipitation |
| Bio 16: Precipitation of wettest quarter | Total precipitation for three months with most precipitation |
| Bio 17: Precipitation of driest quarter | Total precipitation for three months with least precipitation |
| Bio 18: Precipitation of warmest quarter | Total precipitation for three hottest months |
| Bio 19: Precipitation of coldest quarter | Total precipitation for three coldest months |
| Slope | Gradient of land incline |
| Elevation | Elevation in meters |
| Root zone depth | Depth in which crops can extract water and nutrients effectively |
| Available water storage | Amount of water that soil can store for use by crops |
| Taxonomic order | 12 soils orders based on physical, chemical, and biological conditions |

### 2.2. Methods

#### 2.2.1. Constructing and Setting Maxent Model

Modeling corn suitability with Maxent was performed in ArcGIS Pro Version 2.9.0 (Esri, Redlands, CA, USA). In this platform, Maxent requires two inputs to create suitability predictions: (1) point features representing a sample of locations for training the model and (2) point features representing the entirety of the study area. We generated 100,000 randomly distributed points within the CONUS predominant cropland to create a sample for training the model. These points were given attributes indicating crop presence and associated site data (i.e., the 24 bioclimatic, soil, and topography variables). To study the relationships between site variables and crop presence, only the training points are necessary. However, to generate suitability maps, the model needs prediction points representing the entirety of the CONUS cropland at 180-m resolution. Therefore, we generated a point for every cropland pixel at 180-m. This process resulted in approximately 36.7 million points representing 294 million acres of cropland. The prediction points were given the same attributes as the training points (bioclimatic, soil, and topographic).

In addition to these two inputs, Maxent requires two main specifications to run: basis expansion functions and relative weight (RW) of presence to background. Basis expansion functions define how the model will transform the explanatory variables, including linear, quadratic, and product relationships. Hinge as an expansion function is a more recent addition to Maxent that notably improves model performance [48]. We used hinge alongside linear, quadratic, and product functions as expansion functions. Relative weight (RW) defines how the model should interpret background points. When minimized (RW = 1), background points are treated as definite species absence, and when maximized (RW = 100), background points do not factor into the model. When RW is minimized, it indicates high confidence that background points represent absences. In this context in which we are modeling suitability, we cannot be confident that background points are unsuitable, and we also do not want the model to exclude background points (because their lack of corn presence should be meaningful to the model). RW was thus set to 50, allowing absence points to influence the model, but not to the same degree that presence points do. This ratio

allowed the model to learn from background points as areas of possible unsuitability but not rely on them to the extent that it considered them certain unsuitability.

### 2.2.2. Comparing and Evaluating Maxent

Maxent produces a suitability map showing probability of presence (POP) for each 180-m pixel of cropland in the CONUS. Maxent also outputs a receiver operating characteristic (ROC) curve. An ideal ROC curve would be L-shaped and tight against the *y*-axis with rapid maxima, which would make the area under the curve (AUC) reach its maximum possible value of 1.0. The closer the AUC is to 1.0, the higher quality the model. However, Maxent's efficacy can more aptly be evaluated via comparisons with other machine learning methods [37] and existing data on corn production and suitability. In our methods for evaluating Maxent, we made three such comparisons between our Maxent results and (1) those from another correlative machine learning model (RF), (2) existing literature on corn-suitable conditions, and (3) existing corn production data.

#### Comparison with Other Method: Random Forest (RF)

Random Forest (RF) is another correlative machine learning model widely used in landcover classification and other environmental applications [49]. More recently, RF has been applied to crop suitability assessments [50,51]. RF generates decision trees to identify the relationships between presence locations and environmental data [52]. Using iterative subsets of the data, each decision tree casts a vote for the likely classification, and the majority of those classifications determines the model prediction. Like Maxent, RF requires users to set specifications related to size, depth, and number of decision trees that will be used to make predictions. In this case, the model was optimized with 400 trees, 25 leaves, 25% of data available per tree, and 8 randomly sampled variables. We used ArcGIS Presence-only-Prediction and Forest-Based Classification spatial statistics tools for analysis.

We evaluated model performance with statistical metrics and visual inspection to determine which model (Maxent or RF) was better suited for crop suitability assessments. Statistical evaluation involved comparing confusion matrices for the RF and Maxent model outputs. Such confusion matrix comparisons show how accurate each model was at classifying cropland as suitable and unsuitable. Visual evaluation complements this statistical evaluation and can readily show the distribution of these classifications, making it easy to identify issues such as model overfit.

Within confusion matrices, there are several scalar classification metrics that can be used for evaluation and comparison. These metrics are derived from counts of false and true positives and false and true negatives in the classifications. False positives occur when a model wrongly indicates presence where none exists. In most modeling scenarios this would be an error. In this case, however, false positives should not be treated as errors. As mentioned previously, areas of corn absence are not necessarily unsuitable for corn. For example, a hypothetical perfect suitability model would likely classify an area of corn presence as suitable, but it should also classify the adjacent areas (with or without corn presence) as suitable. These areas of suitability and absence would be classified as false positives when they would be evidence of the model behaving correctly. Therefore, the rate of false positives is a misleading variable that has little bearing on the true predictive ability of the model. A low false positive rate, generally an indicator of a good model, in this case may even indicate the model is overfitting to the training data. Given that false positive may be a misleading value, we compared RF and Maxent only though metrics that did not include false positive values in their calculations. Confusion matrices and resulting metrics were calculated using the training data, a sample of 100,000 locations. These metrics include sensitivity (Equation (1)) and Negative Predictive Value (NPV) Equation (2):

Sensitivity is a measure of true positives (corn both present and suitable) out of all real-world presences.

$$\text{Sensitivity} = \text{true positive}/(\text{true positive} + \text{false negative})$$
$$\text{OR} \qquad (1)$$
$$\text{Sensitivity} = [\text{suitable, present}]/[\text{all presences}]$$

NPV measures how much of the land classified as unsuitable were areas of true absence.

$$\text{NPV} = \text{true negative}/(\text{true negative} + \text{false negative})$$
$$\text{OR} \qquad (2)$$
$$\text{NPV} = [\text{unsuitable, absence}]/[\text{all unsuitable}]$$

Higher scores in these two metrics determine which model is more appropriate. Further, comparing the resulting suitability maps generated from Maxent and RF can give a visualized version of a confusion matrix and show the geographic distribution of the area each model classifies as suitable. An output map showing corn-suitable land that lacks gaps and has clearly defined edges would indicate that the model is able to determine suitability without overfitting to the training data provided.

Using Response Curves to Compare to Literature

The Maxent model also produces response curves that plot the values and associated probability of presence for each bioclimatic variable (assuming all others are averaged). Each plot indicates thresholds and ranges where suitability is highest. We compared the response curve plots for growing season bioclimatic indicators (e.g., warmest quarter) with existing literature on suitable corn conditions. Consistency between the thresholds in the response curves and the literature would demonstrate that Maxent successfully predicted the conditions that make an area suitable for corn.

Comparison with Real Cultivation Data

Every five years, the USDA National Agricultural Statistics Service releases its Census of Agriculture, which provides agricultural data at the county level [53]. The most recent census (2017) includes data on corn production and acreage. We compared this real-world data on corn cultivation to our Maxent results, aggregated to the county level. Correlation between probability of presence (Maxent) adjusted for amount of cropland and corn production and acreage statistics were used to determine whether Maxent is an accurate predictor of corn cultivation statistics.

These three comparisons were used to evaluate the model and determine whether Maxent was a superior tool capable of predicting corn statistics and determining thresholds for bioclimatic suitability. After evaluating Maxent's utility and effectiveness via comparisons, we also analyzed bioclimatic statistics in corn-presence areas Maxent classified as unsuitable. When corn grows despite modeled unsuitability, it is an indicator that agricultural management or adaptation techniques are being employed (irrigation, genetic modification, alternations to growing season timeline). To evaluate these regions of unsuitable presence, we classified these locations into nine major regions via multivariate clustering [54]. Clusters were generated with anchoring seeds and the five most important bioclimatic variables (Table 2).

## 3. Results and Discussion

Maxent produces a map of corn suitability across the CONUS, giving a probability of presence score (0–100) for each 180-m pixel of cropland (Figure 1). The map shows suitability is highest in the Corn Belt, the region outlined in red that has historically dominated U.S. corn production [55]. The model classifies 96% of the Corn Belt as suitable for corn (compared to 51% overall in CONUS cropland), and suitability decreases as environmental conditions move from optimal to more unsuitable for corn cultivation. Maxent also yields a

high AUC value of 0.811 (Figure 2), indicating a strong model. However, this metric alone is not enough; comparing Maxent results with that of RF and other data provides more evidence of Maxent's predictive ability.

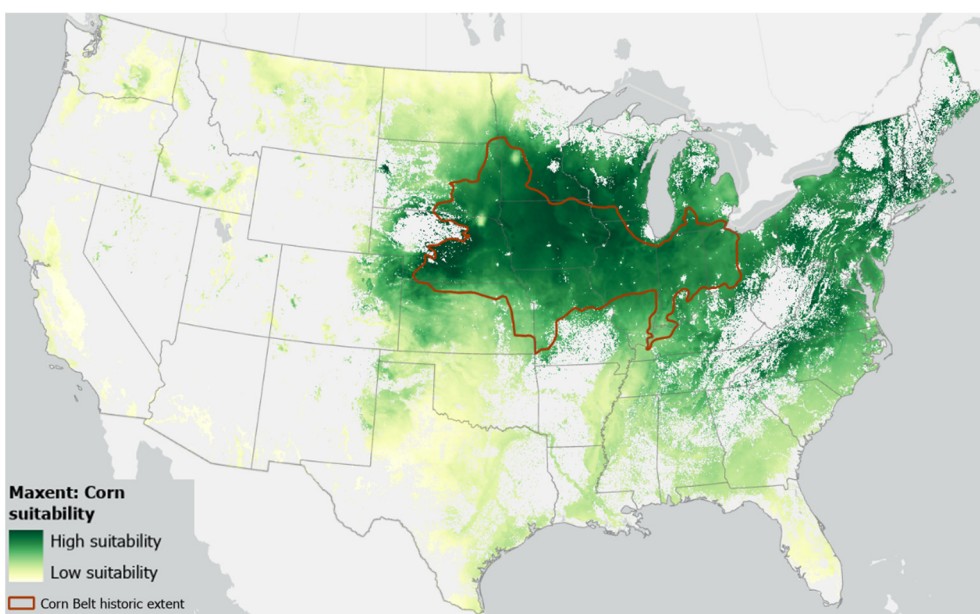

**Figure 1.** Corn suitability in the contiguous United States as modeled. Darker green areas represent greatest predicted corn suitability. Red outline represents the extent of the historic U.S. Corn Belt [55].

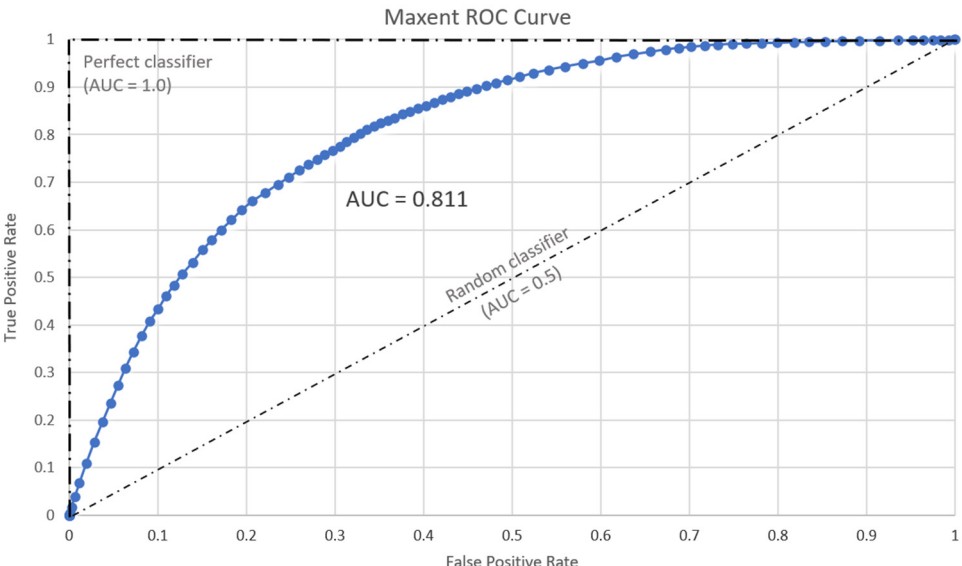

**Figure 2.** Receiver operating characteristic (ROC) curve representing the compromise between true positive rate and false positive rate. A perfect model would have an ROC curve right to the left and top axes, maximizing the area under the curve (AUC) at 1.0. This model has an AUC curve of 0.811, which shows that the model predicts much better than a random classifier.

### 3.1. Model Comparison: Evaluating the Results from RF and Maxent

### 3.1.1. Statistical Evaluation

Figure 3a,b shows the two confusion matrices generated from the corn suitability assessments using Maxent and RF. As mentioned previously, we evaluated the model's ability using sensitivity and NPV because these metrics are not calculated using false positive values. Maxent greatly outperforms RF on sensitivity (0.81 vs. 0.55) and to a lesser extent on NPV (0.88 vs. 0.81). RF's low sensitivity score can be interpreted thus:

RF predicts that of all areas growing corn in the CONUS; approximately half are suitable. Maxent provides a more realistic prediction; 81% of CONUS corn fields are suitable. The RF confusion matrix shows proportionally fewer false positives compared to Maxent. Fewer false positives indicate that RF rarely predicts suitability in areas that do not predominantly grow corn. This suggests that RF is overfitting to match the training data on predominant crop. RF therefore misses many areas that could be suitable for corn but do not happen to be growing corn predominantly over a recent five-year period (due to rotation of presence or other crop). The relatively higher rate of false positives with Maxent indicates that this model is more realistically capturing areas of suitable land growing corn in rotation or not predominantly.

**Figure 3.** (**a**) Maxent and (**b**) Random Forest corn suitability confusion matrices. Sensitivity and NPV are used to compare models since these are the only two metrics computed without using false positive, a misleading metric.

### 3.1.2. Visual Evaluation

Complementing the statistical results, a visual inspection can confirm how well each model captures suitability. Visually comparing the predictions to the predominant crop data can help evaluate whether either model is overfitting to current presence data. The Maxent and RF results were reclassified into grid cells designated as suitable or unsuitable with a threshold of 0.5 for suitability. Figures 4 and 5 combine the model outputs for each model, giving areas of true and false positives and negatives. Areas shaded in green and yellow indicate corn suitability. Green represents suitable areas where corn was also present (true positive). Yellow indicates that the land is cropland suitable for corn, but corn was not present (false positive). Areas in grey represent areas where corn was neither present nor suitable (true negative). Areas in orange represent areas where corn was present but is not suitable (false negative). An accurate model would likely have high incidence of

yellow (false positive) interspersed with green (true positive) because this would indicate that the model is predicting suitability to land without corn but immediately adjacent to corn fields. Large amounts of orange (false negative) would indicate that the model is performing poorly because it is unable to classify significant amount of land cultivating corn as suitable. Orange interspersed with green and yellow would also indicate that the model is overfitting by too closely matching the real-world presence patterns.

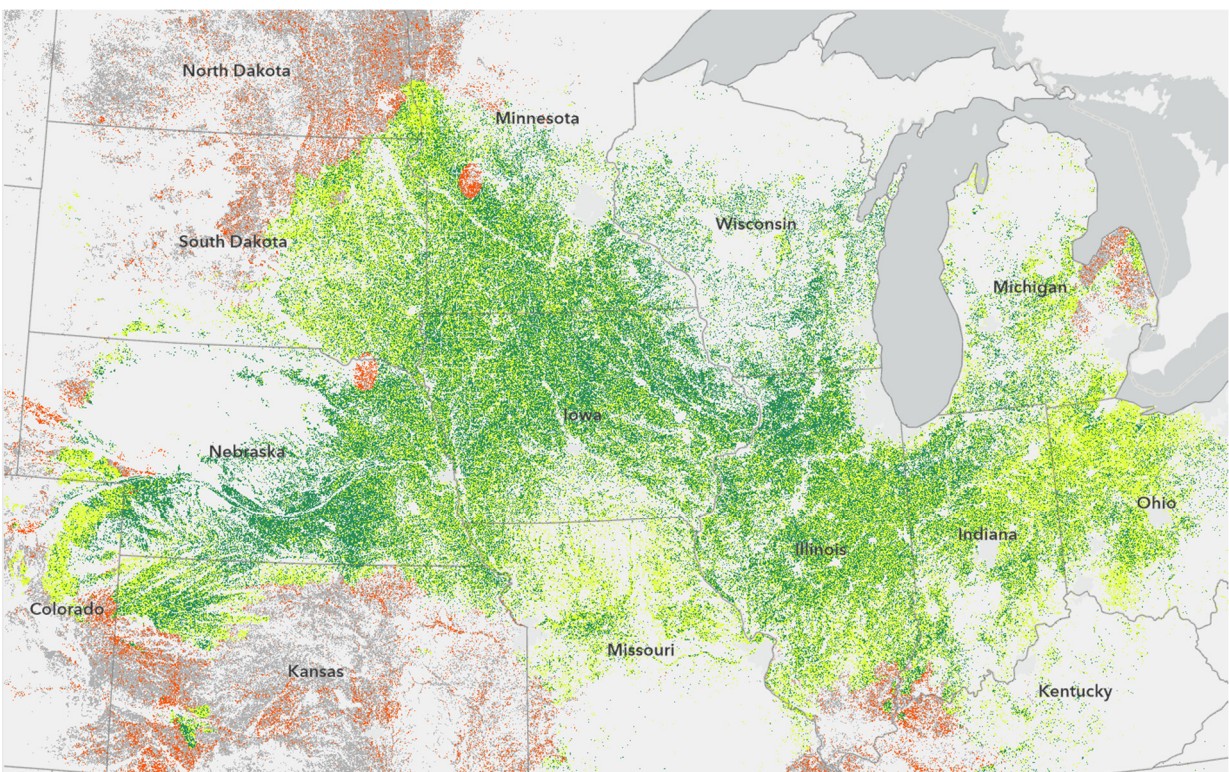

**Figure 4.** Maxent corn suitability compared with real-world presence/absence data. Areas in green represent suitable presence (true positive); areas in yellow represent suitable absence (false positive); areas in orange represent unsuitable presence (false negative); areas in grey present unsuitable absence (true negative). Maxent shows clearly defined areas of corn suitability in and around areas of high corn presence.

In Figure 4, Maxent classifies 49% of all CONUS cropland as suitable for corn. The distribution shows high incidence of interspersed green and yellow, indicating that the model is correctly predicting suitability in areas without corn but with similar site characteristics (bioclimatic, soil, and topographic conditions). More importantly, areas in orange (false negative) are rare (6% of all cropland) and exist at the periphery of suitable land. Overall, this analysis confirms that Maxent can accurately predict suitability. The corresponding RF map (Figure 5) gives strong evidence of overfitting. There is very little yellow in the RF results; RF classifies 6% of cropland as suitable absence (compared to 23% for Maxent), which means the model is underpredicting areas suitable for corn. RF classifies 14% of cropland as false negative (suitable, presence), indicating that the model overpredicted values of the dominant class (absence) at the expense of accurately modeling suitability. The RF model predicts unsuitability even in areas highly likely to be suitable for corn (Indiana) based on suitable environmental conditions and historic extent of the Corn Belt (Figure 1).

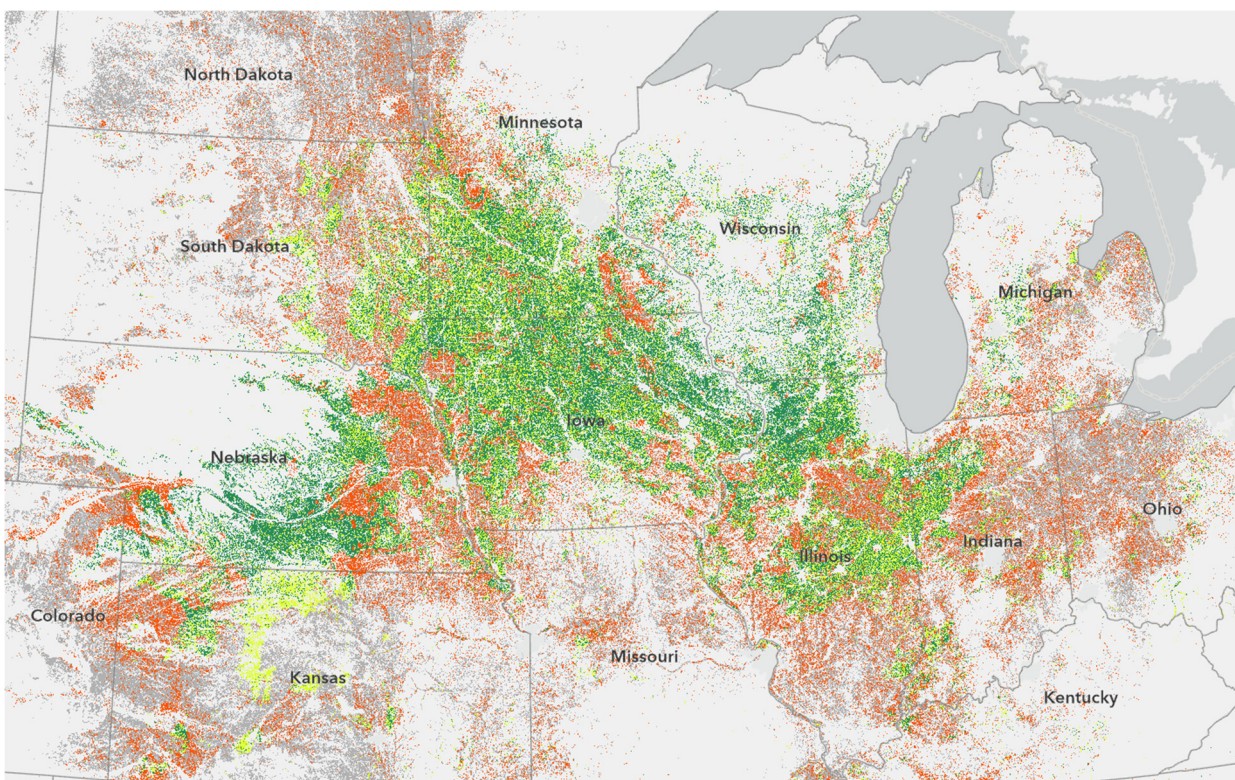

**Figure 5.** Random Forest corn suitability compared with real-world presence/absence data. Areas in green represent suitable presence (true positive); areas in yellow represent suitable absence (false positive); areas in orange represent unsuitable presence (false negative); areas in grey represent unsuitable absence (true negative). RF shows clearly poorly defined areas of suitability with little yellow (false positive), suggesting overfit.

Upon statistical and visual evaluation, Maxent is a more appropriate model for suitability studies. Given the large gaps in the RF suitability maps and its low rate of false negatives (suitable, absence), RF appears to overfit, as indicated by Elith and Graham (2009). The predisposition to overfit does not make RF a poor model, but it does expose a key distinction within species distribution models. One could classify SDMs into two categories: those modeling realized ecological niches (true geographic distribution of a species) and those modeling fundamental niches (areas where said species could be suitable) [56,57]. With its high statistical accuracy (Figure 3) and tendency to overfit, RF belongs to the former and may more accurately predict true species distribution [58]. Conversely, we find that Maxent is more effective at predicting crop habitat suitability (i.e., fundamental niche) [57]. In this application, our objective is finding habitats suitable for corn. Maxent's strong performance in this task suggests that the technique could be applied more broadly to other fundamental niche modeling scenarios.

### 3.2. Comparison with Existing Literature on Suitability Thresholds

Maxent generates response curves for each of the bioclimatic environmental variables (Figure 6) that provide insight into how individual variables impact the model. These curves plot POP against a range of values within each bioclimatic variable (assuming all other bioclimatic variables are set to their respective averages). Overall, the model will predict high suitability in areas where all or most of the bioclimatic variables have their most suitable conditions. Response curves also provide insight into the individual variables and their thresholds. Flatness in response curves indicates that differences in the value of that variable do not have any effect on POP. Conversely, high variation in POP for different values of a variable indicates that that variable contributes valuable information. A higher range in POP means that that variable gives the model more ability to discern between

suitability and unsuitability. Therefore, range in probability of presence (POPmax–POPmin) can be used to determine the relative importance of each variable. Table 2 displays each variable with a POP range above 0.5 ordered by decreasing importance. Variables relating to temperature, specifically extreme and growing season (warmest quarter) temperatures, have the greatest impact on the model.

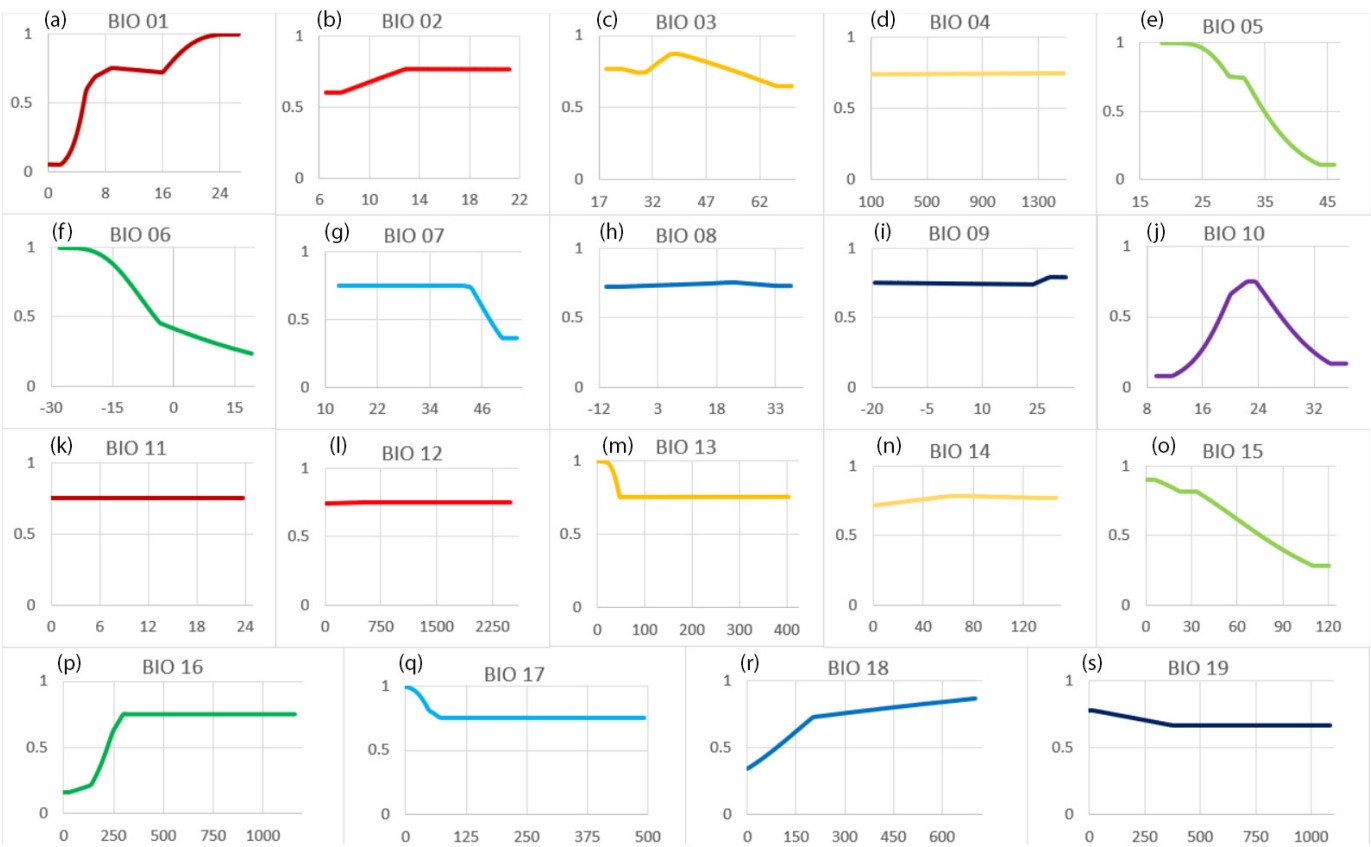

**Figure 6.** (**a**–**s**) Response curves for 19 bioclimatic variables. Response curves show how bioclimatic variable values correspond to suitability thresholds independently assuming all other variables are held static. Individual response curves show the suitable ranges and variation for each variable, and when considered together, suitable areas will be where multiple bioclimatic variables overlap to produce overall suitable conditions.

**Table 2.** Bioclimatic variables with the greatest influence on the Maxent probability of presence (POP) results. Variables are listed in order of descending importance.

| Variable | Name | Range | Max POP | Max POP Value | Min POP | Min POP Value | Trend | Interpretation |
|---|---|---|---|---|---|---|---|---|
| Bio 01 | Annual mean temperature | 0.95 | 1 | >24 | 0.05 | <2 | Positive | Suitability increases with temperature |
| Bio 05 | Max temperature of warmest month | 0.89 | 1 | <28 | 0.11 | >44 | Negative | Suitability decreases in areas prone to extreme summer heat |
| Bio 06 | Min temperature of coldest month | 0.76 | 1 | <−28 | 0.24 | >19 | Negative | High suitability corresponds with cold winters |
| Bio 10 | Mean temperature of warmest quarter | 0.67 | 0.75 | 23 | 0.08 | <12 | Quadratic | Corn prefers warm, not hot, summers |
| Bio 15 | Precipitation seasonality | 0.62 | 0.9 | <7 | 0.28 | >110 | Negative | Suitability increases with low rainfall variation |
| Bio 16 | Precipitation of wettest quarter | 0.58 | 0.75 | >297 | 0.17 | <50 | Positive | Corn prefers medium to high seasonal rainfall |
| Bio 18 | Precipitation of warmest quarter | 0.54 | 0.87 | >700 | 0.33 | <40 | Positive | Corn prefers rainy summers |

Comparing the response curves for the most important bioclimatic variables to existing literature on suitability thresholds can demonstrate Maxent's ability to accurately identify suitable conditions for corn. We selected three growing season (warmest quarter) bioclimatic variables with high importance (Table 1) to compare their response curves with the literature on corn-suitable temperature and precipitation thresholds.

Mean temperature of the warmest quarter has high influence on model decisions (high importance) (Bio10, Table 1). The response curve shows corn is likely suitable (probability of presence > 67%) when mean temperature of the warmest quarter is between 20 and 25 °C (68–77 °F) (Figure 6j). POP peaks at 75% for temperature values 22–24 °C. This finding is consistent with the literature on optimal corn conditions, which estimates optimal mean summer temperature of 22–24 °C [41,59–61].

Maximum temperature of the warmest month (Bio05) is an indicator of extreme high temperatures and heat waves. While corn can tolerate warm to hot summers, it is particularly sensitive to extreme heat [62,63]. The model predicts that corn is most suitable (POP > 75%) when maximum temperature of the warmest month does not exceed 30 °C (Figure 6e). Corn becomes more likely unsuitable (POP < 50%) when maximum temperature of the warmest month reaches or exceeds 35 °C and is damaged by extreme heat of 32–35 °C [41,64]. Extreme mid-summer temperatures are particularly harmful because this timeframe corresponds to the critical tasseling growing stage for corn when it is most vulnerable to heat [62].

Precipitation of the warmest quarter (Bio18) is another indicator during the growing season that gives historic precipitation totals over a three-month period. The model response curve (Figure 6r) shows that suitability generally increases with increased precipitation. According to the U.S. Census of Agriculture, 86% of grain corn acres are rainfed, so summer precipitation is an important variable in corn growth [53]. Corn is likely to be suitable (POP > 75%) when precipitation of the warmest quarter exceeds 278-mm. Below 201-mm, POP declines at a steeper rate to only 50% POP at 89-mm. These various thresholds of suitability are consistent with existing literature on corn's optimal summer precipitation amounts, which describe a positive correlation between growing season precipitation and yield generally across the Eastern U.S. [65]. The area the model classified as suitable to corn receives on average 284-mm of precipitation during the warmest quarter (Figure 1). This finding is consistent with the literature, particularly USDA 1941, which cites approximately 267-mm as the historic optimal precipitation amount for Corn Belt corn production.

For these three critical growing-season bioclimatic variables, the thresholds the model identified are very similar to those found in existing literature on corn suitability. This comparison between model results and literature demonstrates that Maxent is able to successfully identify correlation between corn presence and bioclimatic conditions.

### 3.3. Comparison with Cultivation Data

After establishing that Maxent is a more appropriate model for cropland suitability than RF, we evaluated Maxent's accuracy further by comparing with USDA data on acres harvested and production. Maxent produces a POP score for every 180-m pixel of CONUS cropland. The 2017 Census of Agriculture provides data on corn acres harvested and production in bushels per county. To compare the two, we aggregated the Maxent POP score to the county level. To adjust for counties with high suitability but little cropland, county POP was multiplied by the amount of cropland and then normalized to give a suitability score for each county from 0 to 1, with a score of 1 representing maximized suitability and cropland acres. The values are highly correlated with an $R^2$ of 0.91 (Figure 7), indicating that county suitability score, and by extension, the Maxent suitability model, is a strong predictor of corn area harvested in acres. Maxent results also correlate strongly with production in bushels (Figure 8; $R^2$ = 0.87). This high correlation with observed yield compares favorably against existing literature. Using the non-ML technique APH, Dedeoğlu et al. found an $R^2$ value of 0.83 in comparing modeled wheat suitability and yield [13]. Estes et al. tested multiple SDMs' ability to predict corn yield in South Africa and



found the most accurate model (a generalized additive model) predicted with a correlation of $R^2 = 0.75$ [23]. In their analysis, Maxent had a weak correlation with yield ($R^2 = 0.08$). Therefore, our methods of tuning and setting Maxent give highly accurate results in terms of ability to predict yield when compared with literature. This high correlation demonstrates that these new methods of performing corn suitability analysis contribute novel insight to the field. The model was not explicitly trained on yield data such as production in bushels, but it is a strong predictor of yield, nonetheless. This high correlation between modeled suitability and real-world yield shows the intrinsic link between corn suitability and corn yield and demonstrates the model's predictive power.

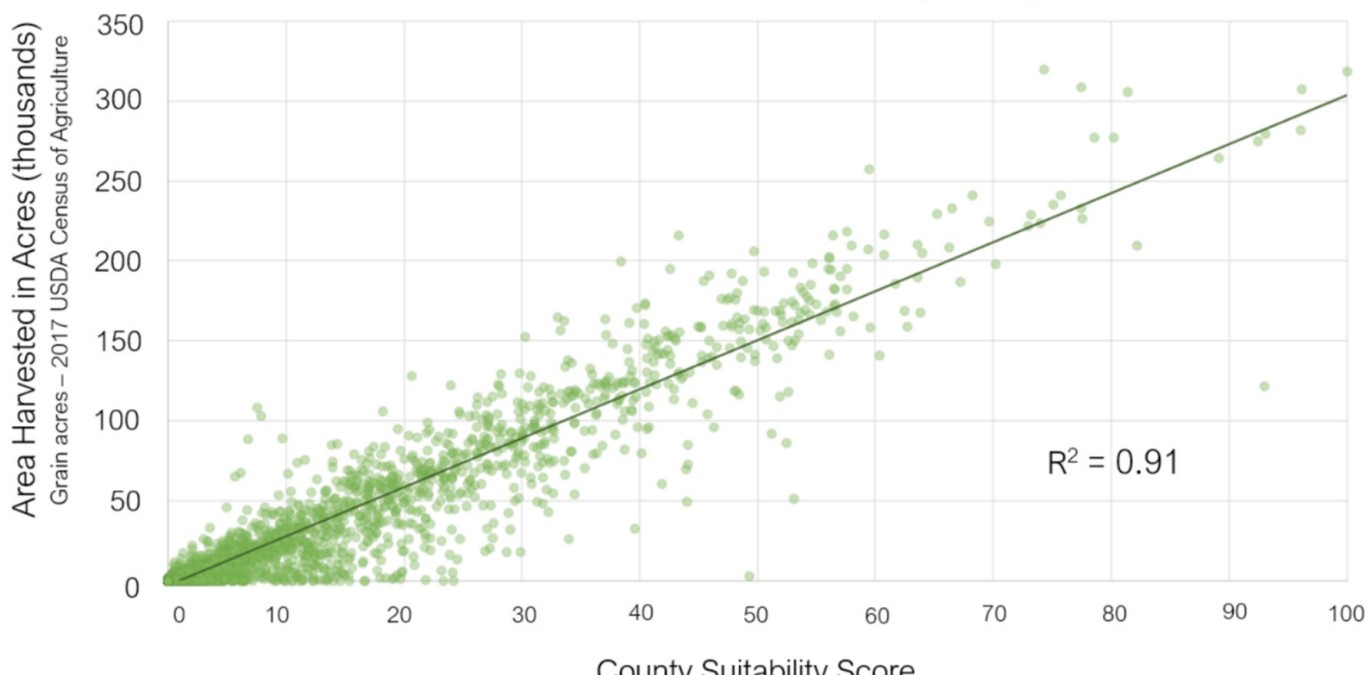

**Figure 7.** Correlation between cropland suitability score and observed acres harvested. Cropland suitability score is probability of presence, aggregated by county, and normalized by cropland area per county in order to adjust for counties with high suitability but little cropland.

Maxent's ability to independently predict corn yield with high correlation shows that intensive mechanistic models are not necessary to make yield predictions. It also implies that, if Maxent were to be run using future climate data, it could predict reasonably accurate future yield predictions, assuming cultivation practices and other socioeconomic factors remain static.

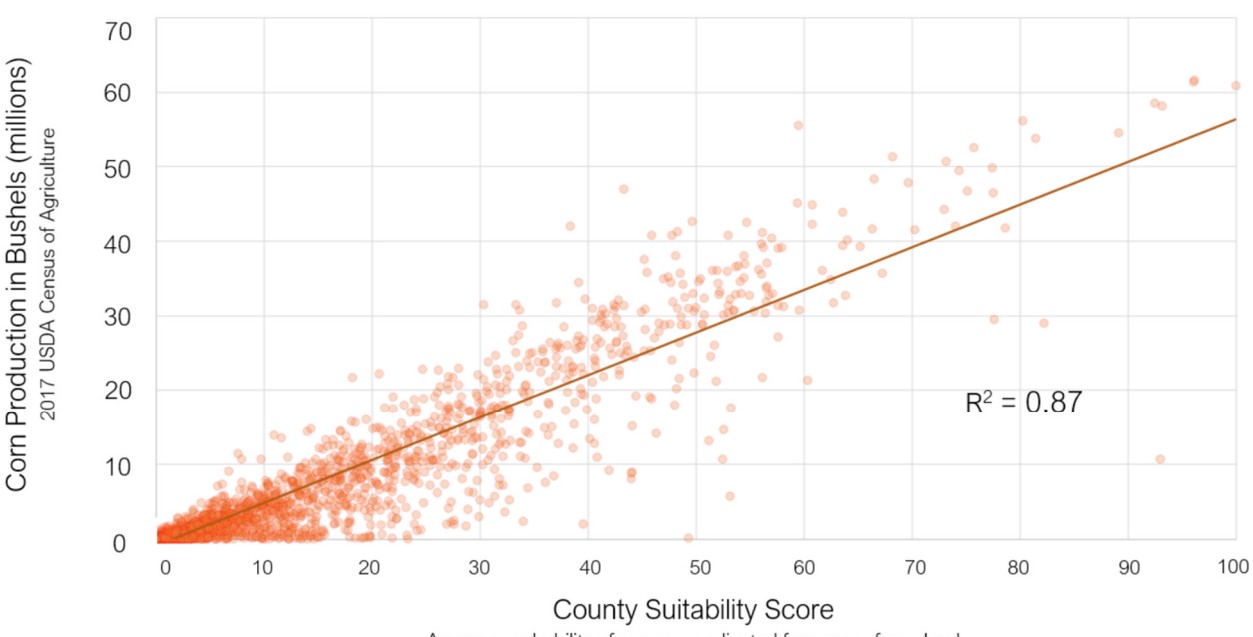

**Figure 8.** Correlation between cropland suitability score and observed corn production in bushels. Cropland suitability score is probability of presence, aggregated by county, and normalized by cropland area per county in order to adjust for counties with high suitability but little cropland.

### 3.4. Application of Maxent: Risk and Necessary Adaptations

Having established the utility and accuracy of Maxent as a method of studying corn suitability and predicting corn yield, we next employed Maxent to identify areas where current corn cultivation is most risky and the need for specific agricultural management or adaptations is greatest. When compared with USDA CDL predominance data [42], the results can show where opportunities and risks exist in current cultivation patterns (Figure 4). Maxent classified 19% of predominant corn cropland as unsuitable. These areas of unsuitable presence (in orange in Figure 4) represent areas with more risk and were further divided into different regions in Figure 9. The fact that corn is cultivated in these areas suggests that other measures are taken to ensure corn production is profitable. Learning exactly how the bioclimate is unsuitable in each region (Figure 9) can explain which adaptations are necessary to cultivate corn in that region.

As previously mentioned, mean temperature of the warmest quarter is a bioclimatic variable with high influence on the model (Table 1). Figure 10 displays the average value of this variable for suitable land (22.1 °C) and areas the model deemed unsuitable but where corn is still observed to grow predominantly. Nearly all of these regions have an average temperature that is at least one standard deviation away from that value in suitable areas. In some cases, such as in corn fields in Texas, mean temperature of the warmest quarter is up to 6 °C higher on average than that of predicted suitable land. The Maxent results show that when mean temperature of the warmest quarter was several degrees above or below that of the range associated with suitable land, these areas may face greater challenges for corn cultivation. These insights suggest that adaptation measures, not explicitly considered by the model, are being taken by growers to ensure that corn continues to grow profitably in areas that may be experiencing excessively warm temperatures. To overcome high heat, several adaptations are available and employed to make unsuitable land more hospitable to corn cultivation. The first is variation in planting and harvesting dates, for which the model does not account but has a strong impact on corn yield [66]. To ensure that the crop hits developmental thresholds like silking, pollination, and maturity during optimal

climate conditions, corn is planted at different points of the growing season depending on the location [67]. Corn planting requires soil temperatures of at least 10 °C, so it is generally planted later in the season at higher latitudes [68,69]. Therefore, planting earlier in the growing season is an important adaptation not considered in the model that may make warmer areas more hospitable to corn. Additionally, Neild and Newman (1990) found that extreme heat threshold is higher for irrigated crops (35 °C) than rainfed crops (32 °C), which suggests that irrigation is another adaptation that could make heat-stressed areas more hospitable to corn.

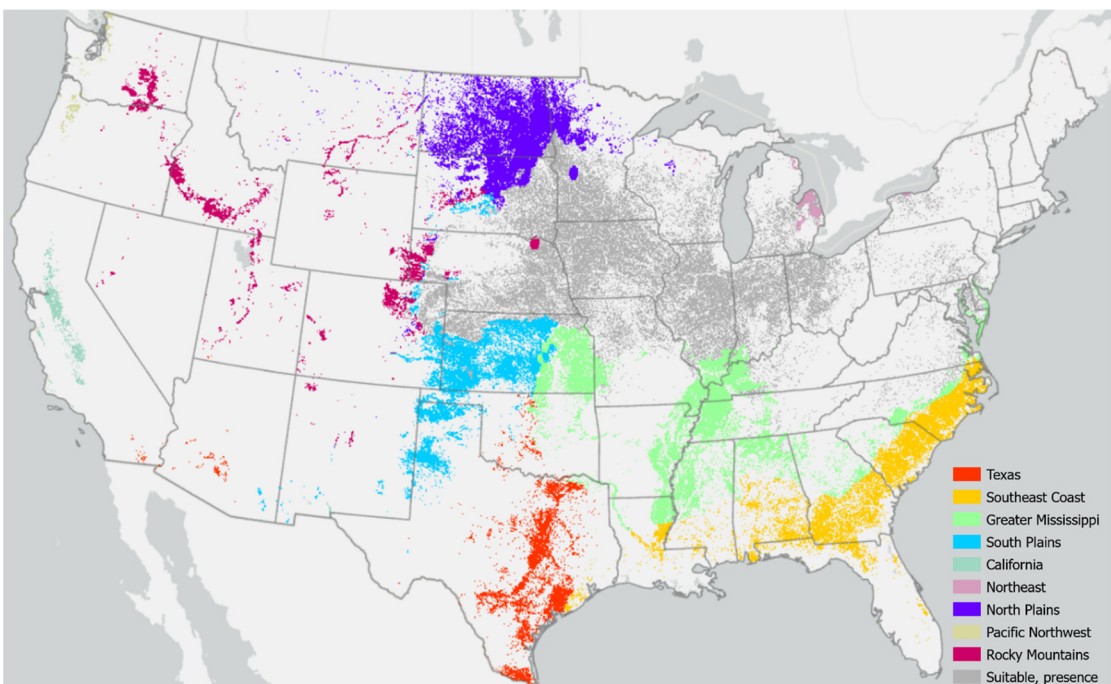

**Figure 9.** Map of regions unsuitable for corn cultivation (unsuitable presence) as predicted by the Maxent model. Areas of unsuitable presence were clustered by geographic proximity and the five most influential bioclimatic variables via multivariate clustering. This process produced nine unique regions of unsuitable presence with unique bioclimatic characteristics. Only clusters with more than 1 million acres were included in Figure 10.

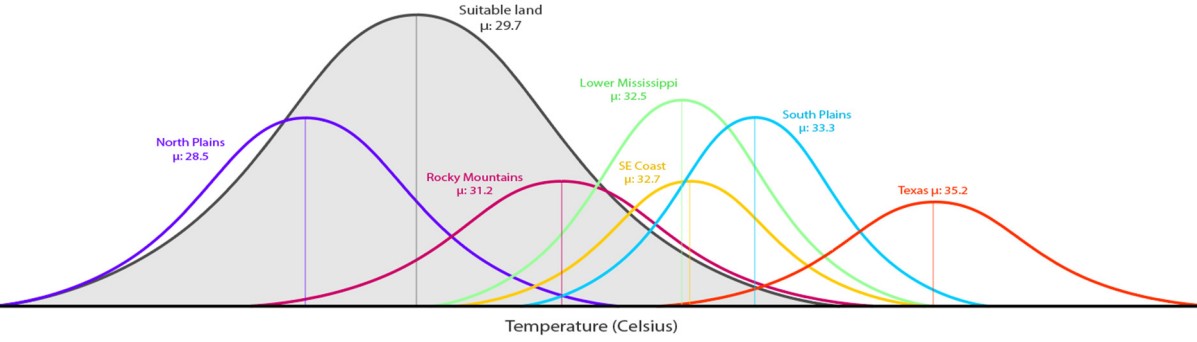

**Figure 10.** Comparison between suitable and unsuitable presence regions for mean temperature of the warmest quarter.

The Maxent model did not take adaptation such as irrigation, genetic modification, or altered growing season timeline into consideration when creating predictions. It can, however, show areas out of suitable bioclimatic ranges where cultivation risk is high and adaptions are or could be beneficial. Diagrams such as Figure 10 can be reproduced for each bioclimatic variable to show exactly the areas at greatest risk and the bioclimatic conditions

that make cultivation difficult. This analysis provides insights into the mitigation strategies needed to overcome bioclimatic risk. The above example demonstrates how Maxent for crop suitability modeling can be used to identify risk and target areas for adaptation.

## 4. Conclusions

With climate change likely to have global impact on agricultural productivity, there is increasing need to understand the relationship between crop cultivation patterns and site characteristics (bioclimatic, soil, and topographic conditions). The purpose of this study was to apply Maxent in studying corn suitability and develop a new set of methods to evaluate such predictions. Maxent was determined to be a more accurate model for cropland suitability estimates in comparison with Random Forest, observed corn cultivation data, and existing literature on suitable conditions. The Maxent model produced a map that shows highest corn suitability in the Midwestern U.S. and identifies areas with more extreme temperatures and precipitation patterns that are associated with increased risk in cultivating corn. The suitability predictions derived from Maxent are also a strong predictor of observed cultivation extent and yield. The ability to predict yield means that complex, data-intensive mechanistic models are not necessary for this type of analysis. The implications of this study are widespread and valuable for the field of land suitability analysis. With evidence that this modeling approach can accurately predict current suitability, it will likely be able to generate suitability predictions under future bioclimatic conditions and show where future climate adaptation and mitigation strategies can be most effective. Given the high correlation with yield, future studies using Maxent set to these specifications may be able to generate future yield prediction under future climate scenarios. Such a research endeavor could yield valuable insight into how climate change could impact U.S. agricultural suitability, extent, and yields. This suitability assessment, therefore, adds to new insights to the field of agricultural modeling and lays the foundation for future cropland suitability assessments to study the impact of climate change on U.S. agriculture.

**Author Contributions:** Conceptualization, A.F., D.P. and D.F.; methodology, A.F. and D.P.; software, A.F. and D.P.; validation, A.F.; formal analysis, A.F. and D.P.; investigation, A.F.; data curation, A.F. and D.P.; writing—original draft preparation, A.F.; writing—review and editing, D.P. and D.F.; visualization, A.F.; supervision, A.F., D.P. and D.F.; project administration, A.F. All authors have read and agreed to the published version of the manuscript.

**Funding:** This research received no external funding.

**Data Availability Statement:** Data on crop land cover is available through the USDA National Agricultural Statistics Service (NASS) at https://nassgeodata.gmu.edu/CropScape/ (accessed on 19 May 2022). Climate data is available through the WorldClim portal: https://www.worldclim.org/data/worldclim21.html (accessed on 18 August 2022). Soil data is sourced from the USDA Natural Resources Conservation Service (NRCS) and is available through this USDA NRCS data portal available for download: https://www.nrcs.usda.gov/wps/portal/nrcs/detail/soils/survey/geo/?cid=nrcseprd1464625 (accessed on 31 January 2022) More information about soil variables and methodologies can be found here: https://www.nrcs.usda.gov/Internet/FSE_DOCUMENTS/nrcs142p2_050734.pdf (accessed on 10 August 2022) Topographic data is derived from multiple agencies including the Airbus, USGS, NGA, NASA, CGIAR, NLS, OS, NMA, among others. Elevation data is available here: https://elevation.arcgis.com/arcgis/rest/services/WorldElevation/Terrain/ImageServer (accessed on 18 August 2022) Slope data is here: https://elevation.arcgis.com/arcgis/rest/services/WorldElevation/Terrain/ImageServer (accessed on 15 August 2022).

**Acknowledgments:** This paper would not have been possible were it not for the dedicated support and insight from talented colleagues. We would like to thank Kevin Butler for providing invaluable advice and wisdom in regard to running, tuning, and interpreting machine learning models; Elvis Takow provided critical feedback and lent industry experience in agronomy and commercial agriculture. Rich Nauman also provided insight into soil science, agriculture, and management, and Michael Dangermond provided assistance with data processing and interpretation of cropland data. Alberto Nieto also provided feedback and insight particularly in interpreting the results and supplemented output from Maxent.

**Conflicts of Interest:** The authors declare no conflict of interest.

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
