# Peer review of "Evaluation of Maximum Entropy (Maxent) Machine Learning Model to Assess Relationships between Climate and Corn Suitability"

_land, doi:10.3390/land11091382_

Round 1
Reviewer 1 Report
The manuscript land-1840273 is well written. The article is good for the journal readers and providing the sufficient information on crop model productivity. The article is suitable for publication in this journal.
Author Response
Dear reviewer,
Thank you very much for taking the time to review this manuscript. I greatly appreciate your positive feedback.
Best,
Abigail Fitzgibbon
Reviewer 2 Report
Demonstrating the effectiveness for assessing relationships between climate and corn suitability by using Maxent method, the study was well written and performed an interesting readability. It employed a coherent and UpToDate references. However, revising the following points will enhance the paper’s outcome.
1. Append a paragraph at the end of Introduction section to clarify the manuscript organization.
2. A descriptive statistics for the dataset is required?
3. Give the maximum entropy model equation?
4. Add the mathematical equation(s) for the scalar classification metrics the author(s) used (i.e. error rate(s), accuracy, …etc).
5. In Fig.3(a&b)>> Is it true? 0.81vs 0.55 OR 0.48 and 0.88 vs 0.78?
6. P#11; L#443 and L# 449>>R2 is R2
I can accept the work after amendment,
Author Response
Dear reviewer,
Thank you for your thoughtful and thorough feedback. I am pleased to hear that you found it well written and readable. As to your specific suggestions, please see the below point-by-point responses:
1. Append a paragraph at the end of Introduction section to clarify the manuscript organization.
Thank you for this suggestion. To provide further clarity on the organization of the manuscript, we have decided to append such a paragraph at the end of the introduction. This paragraph provides a roadmap for the subsequent sections.
2. A descriptive statistics for the dataset is required?
We were not sure to which dataset this suggestion referred. If it refers to the 24 bioclimatic, soil, and topographic datasets included, we thought that descriptive statistics for each of these would take up a great deal of space without being particularly relevant to the modeling and would be inconsistent with how the literature treats input environmental variables. If this suggestions refers to the output prediction dataset, the confusion matrices provide information about about the rates of classification. I hope this adequately addresses this comment.
3. Give the maximum entropy model equation?
Though we would have liked to have included all of the equations associated with Maxent, Maxent is highly complex model, and unlike some other types of modeling, relies on many sets of equations. We instead followed most of the literature and opted to site the original Phillips et al. 2006 paper about Maxent if reader would like to learn more about the equations involved.
4. Add the mathematical equation(s) for the scalar classification metrics the author(s) used (i.e. error rate(s), accuracy, …etc).
While we did include the equations for sensitivity and NPV (page 7), we opted not to include the equations for accuracy, precision, and specificity because we did not use these metrics to evaluate the model. As we explain on page 7, false positive values may not be the best indicator of the model's ability, so we excluded metrics that include false positive values in their calculations. We thought it pertinent only to include equations for those metrics we used to directly compare the models.
5. In Fig.3(a&b)>> Is it true? 0.81vs 0.55 OR 0.48 and 0.88 vs 0.78?
Thank you so much for pointing this out. These were previously values left behind from an earlier draft of the manuscript. The manuscript has been updated with the correct, consistent values.
6. P#11; L#443 and L# 449>>R2 is R2
Thank you for addressing this. The manuscript has been updated with the proper superscripts.
Again, thank you so much for your instructive and thoughtful feedback. It gave us a good bit to think about and certainly improved the quality of the manuscript.
Best,
Abigail Fitzgibbon
Reviewer 3 Report
Dear authors,
thanks for excellent and very interesting paper. No critical comments. Only one idea
2.2.2.1 Chapter seems to be more relevant for results and discussion chapter
Author Response
Dear reviewer,
Thank you so much for your careful consideration of our manuscript. I am pleased to hear that you enjoyed reading it. As to your comment, we have put a lot of consideration into this question below and have decided on the following:
2.2.2.1 Chapter seems to be more relevant for results and discussion chapter
Even before submission we debated whether this section was better suited to the results and discussion section. After consideration, we concluded that it belongs in the methods section. The purpose of this section is to explain how Maxent and Random Forest will be compared. Section 2.2.2.1 introduces and explains the metrics that will be used for this comparison. Since the selection of these metrics is part of the experimental design developed to evaluate Maxent, we believe this section is most suited to methods. However, after further consideration of your comment, we have decided to include a brief section introducing such metrics and Random Forest within the introduction to provide further context.
Again, thank you for your thorough review. Your comment and positive feedback are very much appreciated.
Best,
Abigail Fitzgibbon
Reviewer 4 Report
Authors did great work in the article.
- just one small minor change at pg6 line 24 it should be "Negative Predictive Value (NPV) Eq 2:"
- Can we use advance algorithms such as CNN to assess relationship between climate and corn suitability?
- following papers can be cite for that
@inproceedings{farooq2018weed, title={Weed classification in hyperspectral remote sensing images via deep convolutional neural network}, author={Farooq, Adnan and Hu, Jiankun and Jia, Xiuping}, booktitle={IGARSS 2018-2018 IEEE International Geoscience and Remote Sensing Symposium}, pages={3816--3819}, year={2018}, organization={IEEE} }
Author Response
Dear reviewer,
Thank you for your feedback and careful review of our manuscript. I am pleased to hear that you enjoyed the content. As to your two suggestions, we have the following comments:
1. just one small minor change at pg6 line 24 it should be "Negative Predictive Value (NPV) Eq 2:"
Thank you for pointing this out. This change has been implemented.
2. Can we use advance algorithms such as CNN to assess relationship between climate and corn suitability?
Thank you for bringing up this really interesting branch of modeling. I was not very familiar with convolutional neural networks before, but after reviewing the articles you attached and some other sources, its seems like a fascinating topic. I believe that CNN could potentially used to study current corn suitability. CNN is primarily applied to landcover classification using imagery. I did come across a few applications of CNN specifically for crop predictions.
However, given that CNN relies on classification of imagery, it is only suited to predictions under current conditions. Through this evaluation of Maxent, we hope to develop a method with which future suitability and yield could be modeled under future climate scenarios. This kind of future prediction would not be possible with CNN. Nonetheless, it is a very interesting model and I am glad I got to learn more about it.
Again, thank your for your thoughtful consideration. Your comments were very welcomed and appreciated.
Best,
Abigail
Reviewer 5 Report
About the manuscript with the title "Evaluation of Maximum Entropy (Maxent) machine learning model to assess relationships between climate and corn suitability" I have the following comments:
In the abstract should be highlighted and clearly presented the main gaps in the literature and the principal novelties of the research.
The literature survey about the topics addressed is weak and must be significantly updated.
More information must be provided in the paper about the considered model and how it was executed, as well the other approaches compared.
A discussion section is needed and the conclusions section must be improved with main insights, policy recommendations and practical implications.
Author Response
Dear reviewer,
Thank you for your time and careful consideration of our manuscript. I greatly appreciate the feedback and questions you provided and believe that our thorough review of these suggestions has improved the manuscript. We have meticulously endeavored to provide further clarification and context where requested. Please see below for a point-by-point response to your comments:
1. In the abstract should be highlighted and clearly presented the main gaps in the literature and the principal novelties of the research.
Thank you for providing this comment. After review, we concur that the abstract was lacking presentation of the main gaps in the research. We have worked to more clearly explain these gaps and how this approach is novel. After revision, we explicitly explain how no study before has attempted to apply Maxent to study field crop suitability within the United States. Our approach is a new application of Maxent to this part of this world and this type of species. We also point out that few crop modeling studies have carefully developed model evaluation methods. In this approach, we also develop methods to evaluate the predictive ability of Maxent. We hope that these additions and clarifications to the abstract will provide sufficient explanation of need for and novelty of this approach.
2. The literature survey about the topics addressed is weak and must be significantly updated.
With this feedback, we have decided to provide more context to each component of the literature review in the introduction. We have paid particular attention to articulating the shortcoming of some current modeling methods, the comparison between mechanistic and correlative models and their respective pros and cons, and the value of accurately predicting crop yield. Each section has been augmented and more literature references have been included to the introduction. We believe this makes the literature review more complete and clear in its explanations of why this study is necessary and valuable.
3. More information must be provided in the paper about the considered model and how it was executed, as well the other approaches compared.
We have provided more information about Maxent and correlative species distribution models within the introduction and methods section. Maxent was only directly compared against Random Forest, and we have provided more information about Random Forest throughout as well. Though no other direct comparisons were made between different model results, we also provided more context about other similar models and their applications and applicability to this type of study. We hope that this provides the added context.
A discussion section is needed and the conclusions section must be improved with main insights, policy recommendations and practical implications.
A results and discussion section is included concurrently as we believe that this is the best way to present results consecutively with necessary context interspersed. There have been some notable changes to the results and discussion section to put more emphasis on the novelty of this approach and the value of accurate yield prediction. We have also added to the conclusion section. Though this being a methods paper means that we do not produce specific policy recommendations, we provide firm evidence that Maxent is an effective model for predicting corn suitability and yield. We therefore encourage other studies to employ these methods to broader applications of crop-climate relationships, particularly with future climate predictions.
With this valuable feedback, we have significant sought to clarify, provide context, and streamline the manuscript and believe these revisions have created a strong manuscript.
Thank you very much for your feedback. It is greatly appreciated and very much helped the revision process.
Best,
Abigail Fitzgibbon
Round 2
Reviewer 5 Report
Authors improved the paper.